# Optimal allocation of data across training tasks in meta-learning

## Abstract

Meta-learning models transfer the knowledge acquired from previous tasks to quickly learn new ones. They are tested on benchmarks with a fixed number of data-points for each training task, and this number is usually arbitrary, for example, 5 instances per class in few-shot classification. It is unknown how the performance of meta-learning is affected by the distribution of data across training tasks. Since labelling of data is expensive, finding the optimal allocation of labels across training tasks may reduce costs. Given a fixed budget $b$ of labels to distribute across tasks, should we use a small number of highly labelled tasks, or many tasks with few labels each? In MAML applied to mixed linear regression, we prove that the optimal number of tasks follows the scaling law $\sqrt{b}$. We develop an online algorithm for data allocation across tasks, and show that the same scaling law applies to nonlinear regression. We also show preliminary experiments on few-shot image classification. Our work provides a theoretical guide for allocating labels across tasks in meta-learning, which we believe will prove useful in a large number of applications.

## 1    Introduction

Deep learning (DL) models require a large amount of data in order to perform well, when trained from scratch, but labeling data is expensive and time consuming. An effective approach to avoid the costs of collecting and labeling large amount of data is transfer learning: train a model on one big dataset, or a few related datasets that are already available, and then fine-tune the model on the target dataset, which can be of much smaller size (Donahue et al. (2014)). In this context, there has been a recent surge of interest in the field of *meta-learning*, which is inspired by the ability of humans to *learn how to learn* Hospedales et al. (2020). A model is *meta-trained* on a large number of tasks, each characterized by a small dataset, and *meta-tested* on the target dataset.

The number of data points per task is usually set to an arbitrary number in standard meta-learning benchmarks. For example, in few-shot image classification benchmarks, such as *mini*-ImageNet (Vinyals et al. (2017), Ravi & Larochelle (2017)) and CIFAR-FS (Bertinetto et al. (2019)), this number is usually set to $1$ or $5$. So far, there has not been any reason to optimize this number, as in most circumstances the performance of a model will improve with the number of data points (see Nakkiran et al. (2019) for exceptions). However, if the total number of labels across training tasks is limited, is it better to have a large number of tasks with very small data in each, or a relatively smaller number of highly labelled tasks? Since data-labeling is costly, the answer to this question may inform the design of new meta-learning datasets and benchmarks.

In this work, to our knowledge, we answer this question for the first time, for a specific meta-learning algorithm: MAML (Finn et al. (2017)). We study the problem of optimizing the number of meta-training tasks, with a fixed budget $b$ of total data-points to distribute across tasks. We study the application of MAML to three datasets: mixed linear regression, sinusoid regression, and CIFAR. In the case of mixed linear regression, we derive an approximation for the meta-test loss, and according to which the optimal number of tasks follows the scaling rule $\sqrt{b}$. In order to optimize the number of tasks empirically, we design an algorithm for online allocation of data across training tasks, and we validate the algorithm by performing a grid search over a large set of possible allocations. In summary, our contributions are:

- We introduce and formalize the problem of optimizing data allocation with a fixed budget $b$ in meta-learning.
- We prove that the optimal scaling of the number of tasks is $\sqrt{b}$ in mixed linear regression, and confirm this scaling empirically in nolinear regression.
- We introduce an algorithm for online allocation of data across tasks, to find the optimal number of tasks during meta-training, and validate the algorithm by grid search.
- We perform preliminary experiments on few-shot image classification.

## 2  RELATED WORK

A couple of recent papers investigated a problem similar to ours. In the context of meta-learning and mixed linear regression, Kong et al. (2020) asks whether many tasks with small data can compensate for a lack of tasks with big data. However, they do not address the problem of finding the optimal number of tasks within a fixed budget. The work of Shekhar et al. (2020) studies exactly the problem of allocating a fixed budget of data points, but to the problem of estimating a finite set of discrete distributions, therefore they do not study the meta-learning problem and their data has no labels.

An alternative approach to avoid labelling a large amount of data is *active learning*, where a model learns with fewer labels by accurately selecting which data to learn from (Settles (2010)). In the context of meta-learning, the option of implementing active learning has been considered in a few recent studies (Bachman et al. (2017), Garcia & Bruna (2018), Kim et al. (2018), Finn et al. (2019), Requeima et al. (2020)). However, they considered the active labeling of data within a given task, for the purpose of improving performance in that task only. Instead, we ask how data should be distributed across tasks.

In the context of recommender systems and text classification, a few studies considered whether labeling a data point, within a given task, may increase performance not only in that task but also in all other tasks. This problem has been referred to as *multi-task active learning* (Reichart et al. (2008), Zhang (2010), Saha et al. (2011), Harpale (2012), Fang et al. (2017)), or *multi-domain active learning* (Li et al. (2012), Zhang et al. (2016)). However, none of these studies consider the problem of meta-learning with a fixed budget. A few studies have looked into actively choosing the next task in a sequence of tasks (Ruvolo & Eaton (2013), Pentina et al. (2015), Pentina & Lampert (2017), Sun et al. (2018)), but they do not look at how to distribute data across tasks.

## 3  THE PROBLEM OF DATA ALLOCATION FOR META-LEARNING

In the cross-task setting, we are presented with a hierarchically structured dataset, with task parameters $(\tau^{(i)})_{i=1}^{m}$ sampled from $\mathcal{T} \sim p(\tau)$ and data $(\boldsymbol{x}_j^\tau)_{j=1}^{n_\tau}$ sampled from $\mathcal{D}^\tau := (\mathcal{D}|\mathcal{T}) \sim p(\boldsymbol{x}|\mathcal{T} = \tau)$. Our problem is minimizing the following loss function with respect to a parameter $\boldsymbol{\omega}$:

$$\mathcal{L}(\boldsymbol{\omega}) = \mathbb{E}_{\mathcal{T}} \mathbb{E}_{\mathcal{D}^\tau} \mathcal{L}(\boldsymbol{\omega}; \boldsymbol{x}^\tau) \tag{1}$$

The *empirical risk minimization principle* (see Vapnik (1998)) ensures that the optimum of the empirical risk converges to that of the true risk with an increase in samples from the joint distribution of $(\mathcal{D}, \mathcal{T})$.

### 3.1  META-LEARNING ACROSS TASKS

In the meta-learning problem, we are given the opportunity to adjust the objective function to each task. This adjustment is given by the *adaptation step* of meta-learning (Hospedales et al. (2020)), which represents a transformation on the parameters $\boldsymbol{\omega}$, which is task-dependent and which we refer to as $\boldsymbol{\theta}^\tau(\boldsymbol{\omega})$.

The loss function $\mathcal{L}^{meta}$ is defined as an average across both distribution of tasks and data points. The goal of meta-learning is to minimize the loss function with respect to a vector of meta-parameters $\boldsymbol{\omega}$

$$\mathcal{L}^{meta}(\boldsymbol{\omega}) = \mathbb{E}_{\mathcal{T}} \mathbb{E}_{\mathcal{D}^\tau} \mathcal{L}^\tau(\boldsymbol{\theta}^\tau(\boldsymbol{\omega}); \boldsymbol{x}^\tau) \tag{2}$$

Different meta-learning algorithms correspond to a different choice of $\boldsymbol{\theta}^\tau(\boldsymbol{\omega})$ and we allow for each task $\tau$ to have its own specific loss function. The dependence on $\tau$ built in through the composition with $\boldsymbol{\theta}^\tau(\boldsymbol{\omega})$ makes the loss function a sample from a random function field, when considered as a deterministic function of the data and model parameter $\boldsymbol{\omega}$.

## 3.2 MODEL-AGNOSTIC META-LEARNING

Our case-study for this paper is the meta-learning algorithm MAML developed in Finn et al. (2017). MAML employs a base learner, which parametrizes an estimator family by $\boldsymbol{\omega}$. The algorithm's adaptation step, inspired from *fine-tuning*, performs a fixed number of SGD steps with respect to the data for each task. Thus, the adaptation step maps into the same parameter space as $\boldsymbol{\omega}$.

In MAML with a single gradient step, if we denote the data for task $\tau^{(i)}$ by $(\boldsymbol{x}_j^{(i)})_{j=1}^{n_i}$, and the SGD learning rate by $\alpha$, this transformation is equal to:

$$\boldsymbol{\theta}^{(i)}(\boldsymbol{\omega}) = \boldsymbol{\omega} - \frac{\alpha}{n_i} \sum_{j=1}^{n_i} \nabla_{\boldsymbol{\omega}} \mathcal{L}\big|_{\boldsymbol{\omega}; \boldsymbol{x}_j^{(i)}} \tag{3}$$

This formula corresponds to a full-batch update, employing all the data for task $\tau^{(i)}$, but mini-batch gradient updates can be performed as well. During meta-training, the loss is evaluated on a sample of $m$ tasks, and a sample of validation data points $n_i$ for each task, leading to the following optimization objective:

$$\mathcal{L}^{meta}(\boldsymbol{\omega}) = \frac{1}{m} \sum_{i=1}^{m} \frac{1}{n_i} \sum_{j=1}^{n_i} \mathcal{L}\left(\boldsymbol{\theta}^{(i)}(\boldsymbol{\omega}); \boldsymbol{x}_j^{(i)}\right) \tag{4}$$

During meta-testing, a new (target) task is given and the parameters $\boldsymbol{\theta}$ are learned by a set of target data points following the same equation 3. The final performance of the model is computed on test data of the target task.

## 3.3 DATA ALLOCATION

In this work, we study the problem of finding values of $m$ and $n_i$, such that, under some constraint, the meta-learning loss 2 is minimized given the available data. In this section we make the terms *allocation* and *budget* explicit. The *budget of a meta-learning problem* is defined as the value $b = \sum_{i=1}^{m} n_i$, i.e. the total number of data points available for meta-training. A meta-training set which respects a budget $b$ is a collection of $m$ tasks sampled independently from the task distribution, each composed of a set of $n_i$ data points sampled independently from their respective data-generating distributions, such that $b = \sum_{i=1}^{m} n_i$.

Conversely, given a value of the budget $b$, and a set of tasks $\tau^{(1)}, ... \tau^{(m)}$, a *data-allocation* of the budget $b$ to these tasks is a partition of $b$ into $n_i$ such that $b = \sum_{i=1}^{m} n_i$. If $n = n_i$ for all $i$, this definition is independent of the task samples drawn, and we call this the *uniform* allocation on $m$ tasks for the budget $b$. In this work, we only consider the family of uniform allocations, and we leave the study of non-uniform allocations for future work.

We denote a meta-dataset which respects the uniform allocation with $m$ tasks and $n$ datapoints per task, drawn independently from task distributions $\mathcal{T}$ and data distributions $\mathcal{D}^\tau$ by $\mathcal{M}(\mathcal{T}, \mathcal{D}^\tau; m, n)$. The *optimal uniform data allocation* is given by the values of $m$ and $n$ which minimize the expected test meta-loss (2), after optimizing the meta-parameter $\boldsymbol{\omega}$ on a meta-training set $\mathcal{M}(\mathcal{T}, \mathcal{D}^\tau; m, n)$. Formally, expanding notation to make the dependence of $\mathcal{L}^{meta}$ on $\mathcal{M}$ explicit in its arguments:

$$\boldsymbol{\omega}^*(\mathcal{M}(\mathcal{T}, \mathcal{D}^\tau; m, n)) = \arg\min_{\boldsymbol{\omega}} \mathcal{L}^{meta}(\boldsymbol{\omega}; \mathcal{M}(\mathcal{T}, \mathcal{D}^\tau; m, n)) \tag{5}$$

Then the optimal data allocation is

$$(m^*(b), n^*(b)) = \arg\min_{m \cdot n = b} \mathbb{E}_{\mathcal{T}} \mathbb{E}_{\mathcal{D}^\tau} \mathcal{L}^{meta}(\boldsymbol{\omega}^*(\mathcal{M}(\mathcal{T}, \mathcal{D}^\tau; m, n)); \mathcal{M}^{test}) \tag{6}$$

*Remark.* Notice that in equation (6), $\mathcal{L}^{meta}$ is already an expectation taken over the data distribution which generated the test set. In the definition, this expectation is taken again over the distribution of meta training sets $\mathcal{M}(\mathcal{T}, \mathcal{D}^\tau; m, n)$. It will be useful to write $\mathcal{L}^{meta}(m, n; \mathcal{T}, \mathcal{D}^\tau)$ for the expression minimized in equation 6.

## 4 METHODOLOGY

We develop an experimental framework to recover the optimal data allocation $(m^*(b), n^*(b))$ for a variety of different meta-learning problems.

We consider 3 meta-learning problems in increasing orders of complexity of the data involved:

- A synthetic dataset where each task is a linear regression problem, and the tasks are parametrized by a multidimensional Gaussian. We analyze this setting theoretically, deriving the asymptotics of the optimal allocation with respect to the budget and the learning rate $\alpha$. We then validate our theoretical findings with experiments. The full results are presented in §5.1.

- A synthetic non-linear regression dataset, comprised of 1-dimensional sinusoid regression tasks. Each task is parametrized by amplitude, phase and noise. We perform synthetic experiments meant to validate A, and report results in section §5.2. We compute the allocation curves for multiple budgets, effectively exhausting a portion of the search space for the optimal allocation. We further attempt to recover the optimal allocations by controlling the MAML training regimen via a greedy sequential data allocation. We present results of this method across multiple adaptation step strengths. Details of this procedure in §4.1.

- The CIFAR-FS dataset introduced in Bertinetto et al. (2019). While few-shot classification is not our main focus, we employ this dataset to validate the greedy search procedure in a more realistic context. We rerun the allocation grid search across fewer budgets and compare to the results of the data allocation procedure. We repeat the experiments across multiple adaptation step strengths. Full results can be found in §5.3.

Further details of all computations and experiments are presented in the appendix.

The model family for the final two problems is parametrized by a neural network, whereas for the mixed linear regression problem, exact solutions to each optimization problem are given. The meta-learning algorithm is always MAML. Our first method for exploring the optimal data allocation problem is to compute $\mathcal{L}^{meta}(m, n; \mathcal{T}, \mathcal{D}^\tau)$ as a function of $m$ for various fixed budgets $b$. In conjunction with this, we investigate the possibility of recovering the optimal allocation sequentially, by making incremental data allocation decisions during MAML training time. We proceed to describe this algorithm.

### 4.1 SEQUENTIAL DECISION MAKING FOR THE OPTIMAL DATA ALLOCATION

In this section we present the data allocation problem as a sequential decision making (SDM) process and draw parallels to Shekhar et al. (2020), Lu et al. (2010). Formally, we treat the learning dynamics of the meta-learning algorithm as a Markov Decision Process (MDP), with states given by tuples $(t, \boldsymbol{\omega}(t), \mathcal{M}(\mathcal{T}, \mathcal{D}^\tau; m, n))$, where $t$ denotes the number of gradient updates performed thus far and $\boldsymbol{\omega}(t)$ is the meta-parameter learned at time $t$. An algorithm $\mathcal{A}$ is said to perform *sequential data allocation* if it acts on the state space of the MDP as

$$\mathcal{A}(t, \boldsymbol{\omega}(t), \mathcal{M}(\mathcal{T}, \mathcal{D}^\tau; m, n)) = (t + \Delta t, \boldsymbol{\omega}(t + \Delta t), \mathcal{M}'(\mathcal{T}, \mathcal{D}^\tau; m + \Delta m, n + \Delta n)) \quad (7)$$

where $\Delta t, \Delta n, \Delta m$ are positive integers, $\mathcal{M}'$ is an extension of $\mathcal{M}$ by an i.i.d. sampled quantum of data. Implicit in this formula is the dependence of $\boldsymbol{\omega}$ on $t$, where after the algorithm $\mathcal{A}$ has acted, the parameter $\boldsymbol{\omega}(t)$ has been updated by $\Delta t$ iterations of training on $\mathcal{M}'(\mathcal{T}, \mathcal{D}^\tau; m + \Delta m, n + \Delta n)$.

For our implementation of this algorithm we treat the problem as a contextual bandit with two arms (Lu et al. (2010)). In each state, the action space admits a fixed $\Delta t$, which in our experiments will be an integer between 2 and 100, and $(\Delta m, \Delta n) \in \{(0, 1), (1, 0)\}$.

Our procedure for evaluating the payoff of each decision is as follows:

- Save the meta-parameter $\boldsymbol{\omega}$ at the start of the evaluation procedure.

- Sample $K_0$ extensions $(\mathcal{M}'_k)_{k=1}^{K_0}$ to the meta-dataset $\mathcal{M}$ and according to the distributions $\mathcal{T}$ and $\mathcal{D}^\tau$.

- Compute the loss on each extension after training from the saved meta-parameter $\boldsymbol{\omega}$, for $K_1$ meta-updates, and evaluating the loss on the meta-validation set $\mathcal{M}^{val}$.

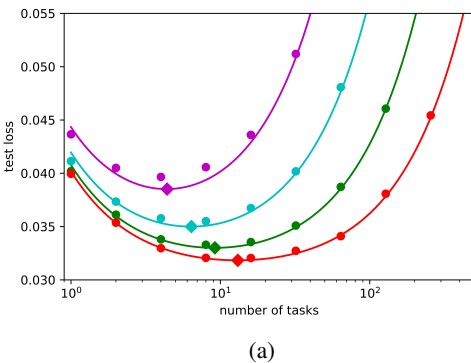 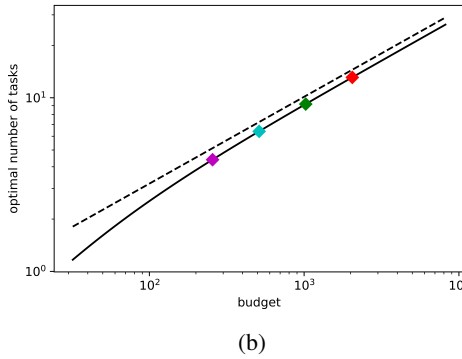

(a) (b)

Figure 1: The optimal number of tasks increases as the square root of the budget in mixed linear regression. a) Average test loss as a function of number of tasks for different values of the budget: magenta - 256, cyan - 512, green - 1024, red - 2048. Circles: average over 100 experiments; Lines: theoretical predictions from equation 8; Diamonds: minima of the loss predicted by theory. b) Optimal number of tasks as a function of the budget, the four minima from panel a are replotted here (diamonds). Full line: theoretical prediction from equation 9; Dashed line: square root law. Hyperparameter values: $p = 32, \sigma = \nu = 0.2, \alpha = 0.3$

As $K_0, K_1 \rightarrow \infty$, the estimation of the payoff from the correct decision converges to the true expected loss of training in the chosen allocation. However, to avoid introducing prohibitively long runtimes, and to respect the budget constraint, in practice, we use small values for $K_0$ and $K_1$.

## 5 RESULTS

### 5.1 SYNTHETIC ANALYSIS: MIXED LINEAR REGRESSION

In mixed linear regression, each task is characterized by a different linear function, and the loss is the mean squared error. The input has dimension $p$ and the output is scalar, therefore the number of parameters of a given task is $p$. The generative model for the parameters is a multivariate Gaussian distribution with $0$ mean and covariance $\nu I$, which we refer to as *task variability*. Furthermore, the output has Gaussian noise added, of width $\sigma$, as in standard linear regression. In the appendix, we calculate the average test loss as a function of $m, b, \alpha, p, \nu, \sigma$. The number of data per task $n$ can be recovered by the substitution $b = nm$. Our result is valid only for a large number of meta-training tasks $m$, and we do not derive error bounds on the approximation. We also simplified the final expression by neglecting powers of $\alpha$ higher than two (see the Appendix for details). The result is

$$\overline{\mathcal{L}}^{test} \simeq \frac{\nu^2}{2} \left[ (1-\alpha)^2 + \alpha^2 \frac{pm}{b} \right] \left( 1 + \frac{1}{m} + \frac{p}{b} \right) + \frac{\sigma^2}{2} \left[ 1 + \frac{p}{b} + \frac{\alpha^2 pm}{b} \left( 1 + \frac{p}{b} + \frac{1}{m} \right) \right] \quad (8)$$

Intuitively, the loss always increases with the output noise $\sigma$ and task variability $\nu$, and decreases with the total budget of data points $b$. Our main focus is to find the optimal number of tasks $m$, for the fixed budget $b$. We find this optimum by differentiating with respect to $m$ and setting the derivative to zero. However, note that errors in the approximation of 8 carry over to the calculation of the optimum. The optimal number of tasks is approximated by

$$m^\star \simeq \frac{1-\alpha}{\alpha} \frac{\sqrt{b}}{\sqrt{p \left( 1 + \frac{p}{b} \right) \left( 1 + \frac{\sigma^2}{\nu^2} \right)}} \quad (9)$$

We note that the optimal number of tasks is of order $\sqrt{b}$. Furthermore, if tasks are all equal ($\nu \rightarrow 0$), then the optimal choice is to allocate all data to a single task (here $m^\star \rightarrow 0$), because a single task is representative of all tasks. On the other hand, the optimal number of tasks is large when the learning rate $\alpha$ is small. In the limit $\alpha \rightarrow 0$, the model ignores the existence of separate tasks and learns a single set of parameters for all tasks. Intuitively, the best choice in that case is to see as many tasks as possible to get a better average over tasks.

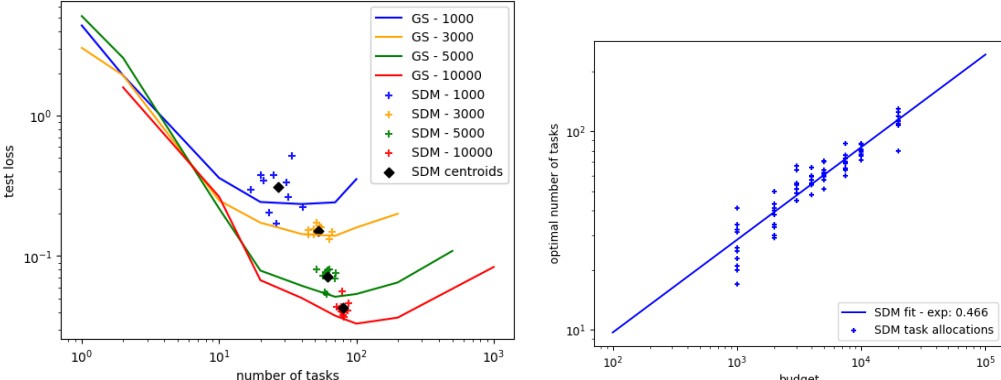

Figure 2: Experimental results on the sinusoid regression dataset. Left: The loss has a unique optimum for the number of tasks, and the optimum increases with the budget size. The SDM algorithm recovers the optimum. Full lines show the loss evaluated by grid search, crosses show the corresponding SDM clusters. Grid search curves are an average over multiple runs. SDM cluster centroids are marked by diamonds. Right: Power law fits for the SDM cluster centroids, optimal number of tasks as a function of budget. The exponent of the power law is $0.466$, close to the predicted square root law.

Figure 1 shows the test loss as a function of the number of tasks, for four different values of budget $b$, and the square root law of the optimal number of tasks as a function of the budget. A good agreement is observed between theory and experiment. Do these results extend to nonlinear problems? In the next sections we test the following two predictions in nonlinear problems: does the optimal number of tasks scales as the square root of the budget? Does the optimal number of tasks increases when decreasing the learning rate $\alpha$?

## 5.2 SYNTHETIC ANALYSIS: SINUSOID REGRESSION

The next experiment involves non-linear regression on a 1-dimensional sinusoid wave dataset. The task distribution is a joint over the distributions of 2 parameters $\tau = (A, \phi)$ where each of $A$, $\phi$, follow predefined distributions (see Appendix) and serve to parametrize the data $(x^\tau, y^\tau)$ by:

$$y^\tau = A \cdot \sin(x^\tau + \phi) \tag{10}$$

MAML is applied to a simple Multi-layer Perceptron base learner, following the architecture in Finn et al. (2017). Details of the MAML training regime for this experiment can be found in the appendix.

Grid search applied to this problem reveals the loss-allocation landscape. Results for various budgets and allocations are depicted in Figure 2. Curves are computed after 1e4-2.5e4 meta-updates, the exact numbers depend on the stopping criterion (see Appendix). To reduce variance in the estimates, up to 10 independent runs from uncorrelated initial conditions are performed for each allocation, and each budget. In the same panel we present the final allocation/performance clusters for multiple runs of the SDM algorithm. The total amount of meta-updates for SDM is commensurate with the grid-search. However, because at the start of training gradient estimates are much noisier, the final performance suffers slightly. Further details of the configuration and training regime for the SDM can be found in 6.

We observe good agreement between the minima of the grid search and the centroid of the SDM allocation cluster. As predicted, we found that the optimal number of tasks increases with the total budget of data points. In Figure 2, we show the optimal number of tasks found by the SDM algorithm as a function of the budget. As predicted by the theory for linear regression, we find a power law with an exponent close to $1/2$ ($0.466$).

In Figure 3 we test the SDM algorithm on 2 different budgets and 3 different learning rates $\alpha$ for the adaptation step of MAML. Theoretically, as $\alpha \to 0$, the optimal allocation should contain an increasing number of tasks, because the meta-learning loss reduces to the cross-task loss 1. As

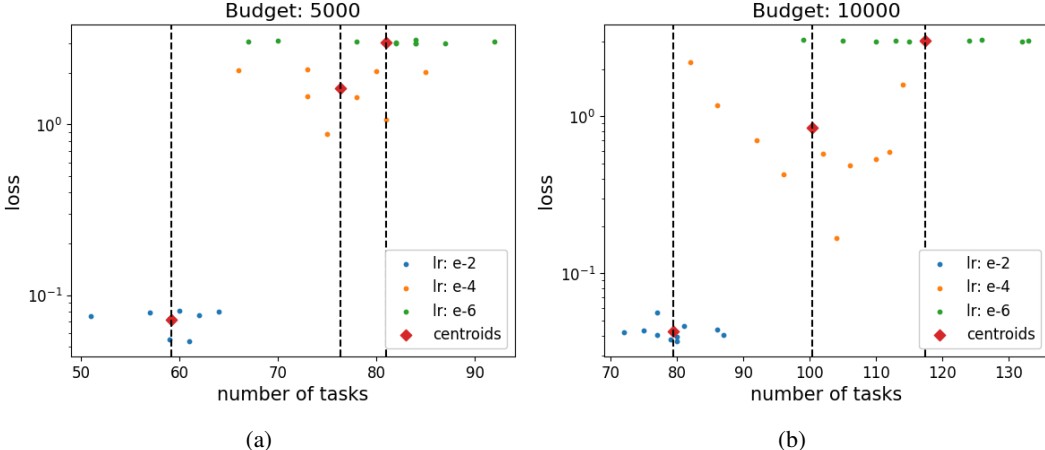

(a)                                                             (b)

Figure 3: The optimal allocation favours more tasks as the learning rate $\alpha$, which controls the strength of the adaptation step, decreases. Correspondingly, performance of the algorithm worsens. Legend: the learning rate values and the coordinate values of the corresponding cluster centroid.

predicted, we observe an increase in the number of tasks recovered by the SDM as $\alpha$ decreases, despite an increase in the variance of the SDM process.

## 5.3 CIFAR-FS RESULTS

In this section we apply our methods to an image classification problem based on the CIFAR-100 dataset. We use the same train / test / validation splits as the CIFAR-FS dataset Bertinetto et al. (2019), but in order to formulate the data allocation problem and give meaningful budget constraints we pre-sample a large set of training and test tasks. The classification problem is 5-way, in that each task contains 5 classes. In CIFAR-FS, the number of data points is fixed to either "1-shot" or "5-shots", meaning 1 or 5 data points per class. In our model, The number of *shots* during training depends on the budget allocation. However, we use a large number of data points for adaptation at testing time in order to provide a minimal variance MC estimate of the meta-loss. Note that in this dataset, the same image may appear in multiple tasks, therefore the total budget does not equal the total number of labels.

In Figure 4a we show the behaviour of $\mathcal{L}^{meta}(m, n; \mathcal{T}, \mathcal{D}^{\tau})$ for $b = 6e4$ and various values of $m$, as well as the results of the SDM algorithm for the same budgets. From the grid search curve, we surmise MAML favours tasks in the optimal allocation, effectively placing the optimum of the curve close to the maximal number of tasks, which is $b/5$ as each task has at least 5 labelled datapoints, one from each class. One possible interpretation of this result is that the model has enough capacity to be able to perfectly cluster 5 classes with a very small number of examples for each class. However, future work is necessary to shed more light on this observation.

The SDM algorithm is unable to recover the optimal allocation set at this extreme point of the curve partly because the process starts at a higher number of datapoints per task than the optimum (to reduce variance in the reward estimates) and partly because the SDM process is likely not given sufficient time between decision ($\Delta t$) and payoff estimation time ($K_1$) to match the large budget (see note in the Appendix). For smaller budgets, however, it becomes more clear that the SDM process favours adding tasks rather than data per task.

In Figure 4b we present the results of the SDM allocation across various smaller budgets for the CIFAR-FS data. A power law fit in this scenario is superfluous, due to the fact that at this end of the budget curves, small changes in $n$ elicit large changes in $m$. However, we present the decision spectrum of the algorithm, which effectively favours adding tasks in proportion of $0.8$ to $0.95$, as opposed to the same scenario in the sinusoid dataset, where the ratios are in between $0.4$ and $0.6$. Note that the initialization of the SDM in this case is at 50 tasks and 50 datapoints per task (10 points for each class). The final column presents the action spectrum of the process for the two extremal

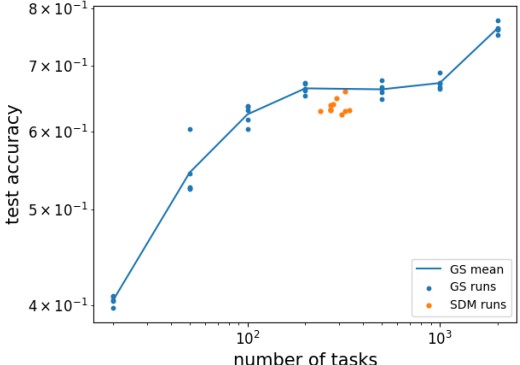

(a) Test accuracy versus number of tasks for a budget of 6e4. Corresponding SDM final allocation cluster.

| Budget | Accuracy | Allocations | Decisions |
|--------|----------|-------------|-----------|
| 10000 | 0.4363 | (200,50) | 0/15 - 0/15 |
| 15000 | 0.4485 | (252,60) | 0/25 - 2/16 |
| 20000 | 0.4634 | (260,76) | 2/23 - 3/20 |
| 30000 | 0.483 | (306,98) | 4/28 - 6/22 |

(b) Table of SDM allocations for smaller budgets averaged over multiple runs. Average final accuracy for each run. Allocations start at (50/50): first coordinate is number of tasks, second coordinate - number of datapoints. The decision column describes the action spectrum for the two extremal SDM allocations at each budget.

Figure 4: Grid search for optimal allocation across a budget of 6e4 on the CIFAR-FS training classes. Various allocations were subjected to multiple runs. We note a steep increase in performance as the number of tasks increases followed by a plateau and second increase as the number of tasks approaches the maximum. The SDM allocations (orange) cluster at the left hand side of the plateau. Table of SDM allocations for smaller budgets.

runs: the one which most favours the decision $(\Delta n, 0)$ and the one which most favours $(0, \Delta m)$. For example, 2/23 signifies that the SDM chose $(\Delta n, 0)$ twice and $(0, \Delta m)$ 23 times.

## 6    DISCUSSION AND FUTURE WORK

In this paper we have analysed the notion of optimal data allocation in meta-learning, by confirming hypothesized properties empirically and proposing methods to efficiently bring meta-learning algorithms close to optimal allocation. In particular, we empirically show that the problem of uniform data allocation has a unique local minimum for each budget. This optimal number of tasks increases with the budget, and we investigate the rate of increase in some specific examples. While it may be that these properties do not hold in general, evidence shows that they do apply in the very simple scenario of linear and non-linear regression, when applied to MAML, where there is more control over the training dynamics. For the CIFAR-FS dataset, a very sparse run of grid search raises questions as to the validity of this assumption, but the variance in the individual meta-training runs is too high to conclusively confirm or disprove this hypothesis.

For the purpose of solving the data allocation problem via sequential decision making, more refined methods can be applied. The SDM algorithm presented in this paper is merely an instance of a policy acting on the state space of the MDP. Designing a more complex state space which takes into account parameters of the MAML training process such as information about previously seen tasks and the model's performance on these tasks, together with an augmented action space can lend solutions of the problem to the framework of reinforcement learning. While preliminary experiments have been performed in this direction, stable enough solutions have not been found, and we defer further exploration.

Overall, our studies exemplify the importance of correct data allocation in meta-learning as well as give a series of preliminary empirical and theoretical insights on the relation between model performance and data allocation for MAML. While the behaviour of other meta-learners need not be in any way similar, we surmise that the problem training models close to optimal allocation remains meaningful, and leave much space for empirical study in a variety of contexts, as well as for the development of a more general theoretical framework.

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
