# OpenReview forum: "Optimal allocation of data across training tasks in meta-learning"
_ICLR.cc/2021/Conference — Reject_

### Official Review · AnonReviewer4 · 2020-10-24
**address an interesting question in meta-learning; uniform simplified assumption; more experiments would be helpful**

**Rating:** 6
**Confidence:** 4

**Review:**

********Summary

In most popular meta-learning approaches, there are usually a pre-defined number of data per task. For example, 1-shot or 5-shot learning. It has shown as the number of data increases in such methods, model performance improves. In this paper, they try to analyze the effect of having different number of tasks with different number of data points in meta-learning benchmarks. Specifically, given a fixed number of data across different tasks as a budget, they want to see if having a large number of tasks with small data in each works better (or worse) than having small number of tasks with more data in each. They focus on MAML as a meta-learning method, and analyzed the results on mixed linear regression, sinusoid regression, and CIFAR. For mixed linear regression, they showed that the optimal number of tasks is \sqrt(b). They also provide an online algorithm for finding optimal number of tasks in meta-learning.

********Positives
- The paper is well-written and well-organized. They explained their approach clearly.

- They tried to answer a very interesting question, that I have always in my mind (and I am sure many other researchers have). It can be generalized to multi-task learning when we have several source tasks for a target task. In multi-tasking there are several sources with various number of data points. The question in this paper can be applied to multi-tasking, but we should consider another important factor, which is task relatedness. Question in that setting might be what is the best combination of tasks with different number of data considering task-relatedness.

********Notes
- There is one confusing part for me. In abstract, they mentioned that "Given a fixed budget b of labels to
distribute across tasks, should we use a small number of highly labelled tasks, or
many tasks with few labels each?", However, 'uniform data allocation' assumption does not align with the original question they wanted to answer! It seems to me that they answer the optimal number of tasks in meta-learning approaches (MAML specifically) when each task has the same number of data.


- I would recommend spending more time on explaining section 4.1. (SEQUENTIAL DECISION MAKING FOR THE OPTIMAL DATA ALLOCATION)

- Where was their theoretical guide as they mentioned in the abstract and introduction, section 5.1?

- Even though the experiments results are really interesting, I would recommend adding one or two more experiments (for example image classification), and also validate their analysis on other meta-learning benchmarks (rather than MAML)

********Reason to accept or reject

They are pointing to very important problem in meta-learning methods. This problem would be interesting to see for other researcher in this area, and it could be generalized to other related research topics such multi-task learning. However, I think they simplified the problem (by uniform assumption), and they could add more experiments to prove their claim. In general, I liked this work, and it could be useful for others to continue since it's not answered yet (at least as far as I know). I would be happier with more experiments or even other meta-learning approaches. So, my score to this paper is marginally above acceptance threshold.

---

> ### Author Response · Authors · 2020-11-13
> **Answer to Reviewer #4**
>
> We thank the reviewer for their comments and the careful analysis of our work. We are grateful for their acknowledgement of our problem formulation, and we would like to address their concerns in what follows.
>
> > There is one confusing part for me. In abstract, they mentioned that "Given a fixed budget b of labels to distribute across tasks, should we use a small number of highly labelled tasks, or many tasks with few labels each?", However, 'uniform data allocation' assumption does not align with the original question they wanted to answer! It seems to me that they answer the optimal number of tasks in meta-learning approaches (MAML specifically) when each task has the same number of data.
> I would recommend spending more time on explaining section 4.1. (SEQUENTIAL DECISION MAKING FOR THE OPTIMAL DATA ALLOCATION)
>
> We agree that the original formulation of the abstract is slightly confusing and will amend our research question to focus exactly on the optimal number of tasks under a budget constraint. We will also make it clear in the abstract that we target uniform allocation in particular in this paper, although the problem formulation remains general enough to encompass both. We also agree that the space devoted to the SDM section is inadequate and will use the extra space for revision to expand this section and give a better description of the implementation. We will also make our code publicly available.
>
> > Where was their theoretical guide as they mentioned in the abstract and introduction, section 5.1?
>
> We thank the reviewer for pointing this out. While it is true that we are unable to make a fully general statement about optimal allocation for meta-learning with MAML, we have seen how in linear /nonlinear regression we observe similar patterns and aim to further consolidate our results in classification tasks. While the solution still seems to be problem dependent, we are confident that once more empirical case studies are conducted, more general patterns will emerge. We will amend our statement for a theoretical guide into a less categorical one, which better suits our partial results in this direction.
>
>
> > Even though the experiments results are really interesting, I would recommend adding one or two more experiments (for example image classification), and also validate their analysis on other meta-learning benchmarks (rather than MAML)
>
> We plan to conduct more experiments as suggested by the reviewer. We hope our current results are enough to motivate the problem and to encourage further experimentation from the community. We strongly believe that the behaviour of optimal data allocation is intimately governed by the function family that the model encodes, and further analysis of other meta-learning frameworks is needed.
>
> Overall, we are grateful for the reviewer's constructive comments which we will incorporate into revisions to our text. We concur that there is strong motivation for presenting this problem to the meta-learning community and are thankful for the reviewer's validation of this opinion.

---

### Official Review · AnonReviewer3 · 2020-10-28
**Review 3**

**Rating:** 4
**Confidence:** 3

**Review:**

Summary
----------

This paper presents an active learning approach to allocation of labels across tasks in meta-learning. The approach is based on a contextual bandit setting, and the approach is tested on linear regression, sinusoidal regression, and a version of CIFAR. The authors also present a theoretical analysis for the linear setting.

Comments
----------

The direction considered in this work is interesting, but the paper feels largely incomplete. Overall, the writing is quite unclear throughout, and the presentation of the SDM algorithm/approach is hard to follow.

The largest shortcoming of the paper is the apparent gap between the the linear results and the nonlinear results. The theoretical analysis for the linear setting is relatively interesting, and the linear experiments appear to match closely. The sinusoidal results also match closely, but this problem can be solved by linear regression on learned nonlinear features, so it is unclear how much of a jump this setting is from the purely linear setting. The proposed approach appears to perform poorly for CIFAR. The authors should include evaluation on standard meta-learning benchmarks such as miniImageNet to show the utility of their approach.

The presentation of the algorithm is quite unclear, and it is unclear how practical the proposed approach is. Formulating the problem as a sequential decision-making problem is logical. However, given the relatively small number of labeled examples considered in this work, the validity of an assumption of a labeled meta-validation set is somewhat questionable. Indeed, given the sparsity of labeled data, it seems unlikely that a large amount of validation data should be withheld from training. I could not find a discussion of the size of this validation set. Indeed, a much more interesting approach for estimating error would be the prediction error associated with adding a new label, as opposed to an external validation set. Moreover, the authors discuss training for 100 epochs after each decision-making step, which seems prohibitively expensive.

Finally, the discussion of existing literature was quite shallow and should be explored much more deeply. In particular, there are likely many connections in active learning and the design of experiments that are highly relevant to this setting.

---

> ### Author Response · Authors · 2020-11-15
> **Answer to Reviewer #3**
>
> We thank the reviewer for their careful consideration of our work. We acknowledge the reviewer's stance that our work could be better explained, and that more experimentation could be performed. However, we would like to provide clarification below with respect to some of the reviewer's comments, which we hope sheds a better light on our work.
>
> > The largest shortcoming […] is the apparent gap between the linear results and the nonlinear results.
>
> We believe the mismatch between results does not point to a "gap", but rather, it is the nature of the problem that different behaviour is exhibited across different datasets and objectives. The distributional assumptions on the data are paramount, as we detail in the following paragraph.
>
> > The sinusoidal results also match closely […] purely linear setting.
>
> We would argue that this setting is a significant jump from the linear problem.
> As stated above, the assumption of the distribution of the data is crucial, more so than the richness of the function family used to perform regression, although both factors contribute. While it is true that the solution retrieved by the neural network may include a set of weights in its final layer that equal the weights found by performing linear regression on the features of the penultimate layer, it is impossible to determine what the distribution of these features is, once passed through the neural network nonlinearities.
> Hence the results of mixed linear regression do not generalize directly.
> Additionally, to motivate the importance of the distributional assumption, it is useful to contrast the scenarios in which the task distribution is uniform over some task generating parameter, as opposed to being close to a delta distribution centred on a particular value of this parameter. Even if the model has enough capacity and benefits from a sophisticated enough training routine to perfectly fit all tasks across both scenarios, it is clear that in the second scenario all that is needed is to increase the number of samples in the most likely task (unlikely tasks effectively don't matter in the expectation of performance), whereas in the first scenario, the answer depends on the similarity between tasks and the sample efficiency of the algorithm. Despite all tasks and the function family of the model being the same, different distributions on the task space entail different optimal allocations.
>
> > The proposed approach appears to perform poorly for CIFAR.
>
> While upon visual inspection, the SDM seems to stop short of the ground truth optimal number of tasks, the authors explain in detail that this is due to the initialization of the algorithm and that the proportion of correct decisions performed by the SDM is very high, around 90% across multiple runs.
> Furthermore, if the comment refers to the qualitative mismatch between the CIFAR results and the regression results, we will clarify in the text that overlap in these results was never part of our expectation.
>
> > The authors should include evaluation on standard meta-learning benchmarks
>
> We plan to produce results for other classification problems, and we thank the reviewer for the suggestion of miniImageNet. We would argue that CIFAR-FS is itself a standard benchmark and in no way a trivial problem for meta-learning. Bertinetto et.al (2019) explains the rationale behind its introduction.
>
> > The presentation of the algorithm is quite unclear, and it is unclear how practical the proposed approach is.
>
> We will address this in the paper revision by including a more detailed description of the algorithm.
>
> > Finally, the discussion of existing literature was quite shallow […] there are likely many connections in active learning and the design of experiments.
>
> Indeed, there are many interesting connections which are not explored in the paper. However, our paper contains no less than 4 references to papers that implement active learning methods, as well as a reference to a standard survey that can guide the reader's exploration of the field (this is already present in section 2.1). We would be glad to incorporate a discussion of any references proposed in subsequent revisions.
>
> Overall, we agree with the reviewer that the problem requires further study. The fact that it deserves further exploration, and that it is a timely and interesting problem to the research community in meta-learning motivated us to present the current version of the results, which we believe are telling of the difficulty an intricacy of the problem as well as exponentiates some of the methods that render it tractable. While we nowhere claim that we provide a complete solution, we feel our work publicises an important direction for research and provides a non-trivial amount of insight. We hope that the reviewer will consider amending their score subject to revisions addressing their comments.

---

### Official Review · AnonReviewer1 · 2020-10-28
**The problem needs to be further studied**

**Rating:** 4
**Confidence:** 3

**Review:**

The authors study the problem of finding the optimal allocation of labels across training tasks given a fixed budget. The authors mainly want to answer the question that "if the total number of labels across training tasks is limited, it is better to have a large number of tasks  with very small data in each or a relatively smaller number of highly labeled tasks?" Although the authors prove that the optimal scaling of the number of tasks in synthetic tasks, the conclusions remain unclear while handling real-world challenging tasks. More specifically, the authors get the result on CIFAR-FS that the optimum allocation is close to the maximal number of tasks, which is weird and not consistent with previous studies in the paper.

Although the authors provide a possible interpretation that the model has enough capacity to perfectly cluster 5 classes with a very small number of examples for each class, such an interpretation is not convincing considering that meta-learning models still have a big room to improve. Moreover,  most meta-learning models adopt a similar or even more complex model, the key research question proposed in the paper about optimal allocation between tasks and labels will be less interesting if the proposed interpretation is correct.

Overall, the question proposed in the paper is interesting and important. I will suggest that the authors should further study the weird outcome on the real-world datasets.

---

> ### Author Response · Authors · 2020-11-13
> **Answer to Reviewer #1**
>
> We thank the reviewer for their consideration of our work. We would like to discuss the reviewer's main concern related to the mismatch between the results on synthetic vs natural data below.
>
> > More specifically, the authors get the result on CIFAR-FS that the optimum allocation is close to the maximal number of tasks, which is weird and not consistent with previous studies in the paper.
>
> We agree with this statement, and believe it warrants further investigation on other classification tasks. However, we would like to point out that we nowhere claim that the behaviour of the simple regression studies extends beyond this context. Indeed, we believe there are combinations of meta-data distributions and model families where the answer can be that picking a single task with as much data as possible is most effective, or, on the other hand, picking as many tasks as possible with a single datapoint in each. The conclusion of the paper should not be interpreted as a solution to the problem of data allocation, but merely as a forerunner to more research and a statement that this problem is complex enough even under simplifying assumptions to warrant study. Furthermore, we hope that our problem formulation and our method of casting the problem as a SDM process (which draws inspiration from active learning - and should be familiar to the community) can serve as a guide to future research on the topic. We would like to stress that results on CIFAR-FS are definitive as presented. The behaviour observed is not the result of a shortcoming in methodology - it is the ground truth of the problem. The mesh of the grid search can always be reduced, and more runs can eliminate some of the variance in the results, but other than these considerations, we are certain of the results we see in this case. Moreover, arguments can be made about the SDM process and how appropriate it is in this case, what combinations of hyperparamters work best and how the problem depends on these choices; our aim was not to give a universal solution to the problem, but rather to present the interesting behaviour of MAML on this dataset as well as the behaviour of the SDM procedure, which was validated on synthetic data, which imposes a completely different training dynamics on the model. This was done without overtuning and complex hyperparameter choices, but with rather naturally selected run-time configurations, which are detailed in the appendix. For this case, the variance of the final result depended on the initial performance estimates, for which around 40-50 points per task were necessary to achieve stability. We describe in our results section that once the SDM is given this initial configuration, most of its choices are adding tasks and hence the final result should be regarded as commensurate with that of the grid search given the limitations.
>
> > Although the authors provide a possible interpretation that the model has enough capacity to perfectly cluster 5 classes with a very small number of examples for each class, such an interpretation is not convincing considering that meta-learning models still have a big room to improve. Moreover, most meta-learning models adopt a similar or even more complex model, the key research question proposed in the paper about optimal allocation between tasks and labels will be less interesting if the proposed interpretation is correct.
>
> We agree with this statement completely and agree that our interpretation was presented without sufficient caveats and appears too categorical in the text. We would like to point out that this does not throw doubt over the significance of the problem, but merely presents a setting in which we observe discordant results from those of the synthetic examples. Whether this is due to the nature of classification tasks is merely speculation and we will modify the text to reflect this. Moreover, how much this result generalizes beyond this setting is still unclear and we will perform further experimentation.
>
>
> Overall, we agree with the reviewer's evaluation that more work is needed, but we find our initial results to be detailed enough to hold good value to the community. It is unlikely that one or two extra empirical studies will shed complete light on the problem, and we consider that the paper paints an interesting picture of the general interaction between meta-learning models and meta-data in its current form. Since the submission we have obtained further results, but would be very interested in the reviewer's opinion on how to make our presentation more complete.

---

### Official Review · AnonReviewer2 · 2020-10-30
**Review #2**

**Rating:** 4
**Confidence:** 4

**Review:**

This paper proposes a data allocation scheme for meta-learning. The authors argue that it is important to consider the number of total tasks versus the number of datapoints per task given a fixed budget of the total number of datapoints since labeling is expensive for large datasets. The paper presents an algorithm that does the data allocation using a sequential decision process (SDM) and models the problem as a two-armed contextual bandit where for each fixed number of iterations, the agent can choose to add 1 task or 1 datapoint per task. There are some theoretical results on the optimal data allocation in the linear regression setting. The authors also empirically show that the number of the fixed budget and the data allocation can affect performance fairly a bit in a sinusoid regression dataset and the CIFAR-FS dataset. The proposed SDM algorithm can also recover the optimal data allocation in the sinusoid regression setting.

For the pros of this paper, I think the idea of better data allocation for meta-learning is novel since, in standard meta-learning settings, we usually assume some predefined number of data per task and number of tasks. I also like the analysis of the data allocation scheme in the linear regression setting, which sheds some light on how we should balance the number of tasks and the number of data per task.

Meanwhile, for the cons of this paper, I can't really see the significance of the problem of data allocation, particularly the data allocation set-up in this paper where each task has the same number of datapoints. Why is the problem important? I don't find the argument about the limited budget in the paper convincing enough since it seems that some naive, hand-designed data allocation schemes (e.g. having as many tasks as possible) work just well as shown in the experiments regardless of the value of the budgets. Is this problem a bit contrived and not very important in the field of meta-learning research? I believe the setting where we do not have uniform data allocation is much more important since some tasks might be much harder to get labels, but this is not explored in detail in this paper.

Moreover, it seems that the initialization of the data allocation of the proposed SDM algorithm matters a lot. If the algorithm is so brittle w.r.t. the initialization, why can't we just do grid search instead? Also, in the CIFAR-FS results, the optimal data allocation discovered by the grid search is simply using as many tasks as possible, which is what people are doing in practice. Such results make me doubt the significance of the problem setting again.

Finally, the paper is not evaluated on some widely used meta-learning benchmarks such as Omniglot, miniImagenet and etc.. Getting more empirical evidence on those benchmarks would be important.

Given the above comments, I would vote for a reject for this paper for now.

---

> ### Author Response · Authors · 2020-11-15
> **Answer to Reviewer #2**
>
> We thank the reviewer for his careful consideration of our paper and for comments and suggestions. We would like to address some of the reviewer's concerns below, which we feel paint the paper and its results in a somewhat negative light.
>
> > I can't really see the significance of the problem […]
>
> We appreciate the statement that "the idea of better data allocation for meta-learning is novel". Subsequently, we believe the comment above is slightly uncharitable. As we proceed to show both in synthetic and empirical studies, the problem of uniform allocation is itself non-trivial, with results being dependent on the real data distribution and the model's capacity. In certain synthetic cases, this problem is solvable as we attempt to show.
> As to its relevance, we believe the problem of uniform allocation is an important precursor to the non-uniform case which cannot be benchmarked against simple methods e.g. grid search. Furthermore, it seems unlikely for any algorithm proposed to solve non-uniform allocation to produce credible solutions if it is unable to pass acceptable benchmarks on the uniform problem.
>
> > I don't find the argument about the limited budget […] convincing […].
>
> We attempt to show that it is not indeed true that a randomly chosen data allocation performs just as well as the optimal one, in general. Our results highlight this behaviour clearly.
> On the other hand, in order to justify the budget constraint, one can look at the non-uniform case where the budget has a natural definition as the total number of data samples across all tasks. We agree that for tasks such as classification the total number of labelled samples (with the relevant cross-task overlap) is a more natural constraint. However, this constraint does not generalize directly to all meta-learning problems whereas ours does. Furthermore, it provides a universal point of comparison between problems.
>
> > Is this problem a bit contrived and not very important […]?
>
> We believe the problem is an abstract engineered setting where meaningful results can be given. We agree with the reviewer that for practical considerations better problem dependent formulations may exist; tackling these was not the aim of our paper.
>
> > I believe [non]-uniform data allocation is much more important […].
>
> We agree and are currently working towards results in the non-uniform setting. However, given the length and scale of the problem, as well as the necessary motivational background to introduce it, we believe that a single paper describing the results of both problems would be overly dense and fail to shed enough light on the subtleties of the problem and the difficulties of the approach.
>
> > If the algorithm is so brittle […] just do grid search instead?
>
> The initialization of the SDM matters in as much as there is enough initial data to produce acceptably low variance estimates of the model performance. Smaller initializations were considered, but the variance in the final allocation naturally increases. We chose to present the results that achieve a reasonable stability threshold.
>
> Performing grid search with a fine enough sweep, for large budgets and complex models is prohibitive. We will add wall-times for the SDM algorithm contrasting them to those for grid search (< 1 day vs. >  5 days). Furthermore, the problem of variance / brittleness is not estranged to grid search which requires averaging over many runs to reduce the variance in each estimate.
> Moreover, the SDM algorithm runs online with the training of MAML and hence produces a reasonable estimate on O(1) (MAML run time) for any budget.
>
> > CIFAR-FS results […] doubt the significance of the problem setting
>
> We agree with the reviewer that in practice, one would use all the data (=labelled examples) available, but it is impossible to use all the tasks available, which for CIFAR-FS amount to 100 choose 5 \simeq 75e6 tasks. Furthermore, the budget constraint is not an active consideration for practitioners when training from a static database of labelled samples, but it is indeed relevant for practical applications which receive data online or for which requesting extra data incurs a cost.
>
> > [The] paper is not evaluated on […] Omniglot, miniImagenet.
>
> We agree and will present results on miniImageNet in further studies. However, for Omniglot one does not have sufficient samples per class in order to make the allocation problem sufficiently diverse; furthermore, as past studies show, Omniglot is a comparatively easier scenario for meta-learning than CIFAR-FS; we believe results on the latter are more relevant to our question.
>
> In general, we share the reviewer's concerns, but we believe our results are interesting in this setting and meaningful enough to draw attention to the importance of the problem of data allocation in meta-learning. We hope that the reviewer will consider amending their score in light of our explanations subsequent to our revision of the paper.

---

### Decision · Program_Chairs · 2021-01-07
**Final Decision**

**Decision:**

Reject

**Comment:**

This paper studies the problem of how data should be balanced among a set of tasks within meta-learning. This problem is interesting, and largely hasn't been studied before. However, the reviewers raised several shortcomings of the current version of the paper, including the significance of the problem setting, the limited experimental study (i.e. the only experiment with real data is CIFAR-FS), the depth of the related work section, and the clarity/impreciseness of the writing. Further, the paper has not been revised to address any of these shortcomings. As such, the paper is not ready for publication at ICLR.